# Apolipoprotein-L1 (APOL1): From Sleeping Sickness to Kidney Disease

**DOI:** 10.3390/cells13201738

**Published:** 2024-10-20

**Authors:** Etienne Pays

**Affiliations:** Laboratory of Molecular Parasitology, Institut de Biologie et de Médecine Moléculaires (IBMM), Université Libre de Bruxelles, 6041 Gosselies, Belgium; etienne.pays@gmail.com

**Keywords:** sleeping sickness, chronic kidney disease, INF-I inflammation, APOL1 risk variants, N264K variant, membrane dynamics, apoptosis, mitochondrion fission, mitophagy, cholesterol microdomains, cation channels, PI4KB, NCS1, ARF1

## Abstract

Apolipoprotein-L1 (APOL1) is a membrane-interacting protein induced by inflammation, which confers human resistance to infection by African trypanosomes. APOL1 kills *Trypanosoma brucei* through induction of apoptotic-like parasite death, but two *T. brucei* clones acquired resistance to APOL1, allowing them to cause sleeping sickness. An APOL1 C-terminal sequence alteration, such as occurs in natural West African variants G1 and G2, restored human resistance to these clones. However, APOL1 unfolding induced by G1 or G2 mutations enhances protein hydrophobicity, resulting in kidney podocyte dysfunctions affecting renal filtration. The mechanism involved in these dysfunctions is debated. The ability of APOL1 to generate ion pores in trypanosome intracellular membranes or in synthetic membranes was provided as an explanation. However, transmembrane insertion of APOL1 strictly depends on acidic conditions, and podocyte cytopathology mainly results from secreted APOL1 activity on the plasma membrane, which occurs under non-acidic conditions. In this review, I argue that besides inactivation of APOL3 functions in membrane dynamics (fission and fusion), APOL1 variants induce inflammation-linked podocyte toxicity not through pore formation, but through plasma membrane disturbance resulting from increased interaction with cholesterol, which enhances cation channels activity. A natural mutation in the membrane-interacting domain (N264K) abrogates variant APOL1 toxicity at the expense of slightly increased sensitivity to trypanosomes, further illustrating the continuous mutual adaptation between host and parasite.

## 1. Function of the Extracellular APOL1 Isoform: Trypanosome Lysis

### 1.1. APOL1 Secretion: A Recent Invention for African Trypanosome Killing

Apolipoprotein-L1 (APOL1) is the primate-specific member of a dynamic family of mammalian proteins involved in several diseases, but with largely unknown mechanisms [1]. This protein was discovered as a minor constituent of the densest fraction of serum High-Density Lipoprotein particles (HDL-C) involved in cholesterol recycling in the bloodstream (“good” cholesterol) [2]. As far as we know, APOL1 is the only secreted protein among numerous APOL family members, whether in primates or in rodents, meaning that the original functions of APOLs must be intracellular. Even in the case of APOL1, such functions could still exist, because alternative transcript processing generates both intracellular and extracellular isoforms.

APOL1 expression is strongly increased under inflammatory conditions linked to viral or parasite infection [3,4], and APOL evolution in primates suggests a role in resistance to pathogens [5]. Accordingly, the secreted APOL1 isoform was found responsible for innate human resistance to infection by the prototype African trypanosome *Trypanosoma brucei brucei* [6], which causes sleeping sickness, a lethal inflammatory parasitic disease widespread in the sub-Saharan part of the continent [7]. Given the African origin of human ancestors and related primates, the discovery of such isoform, which allowed resistance to deadly parasites, may have been linked to the success of human evolution [8]. However, two *T. brucei* subspecies, named *T. b. rhodesiense* and *T. b. gambiense*, managed to resist APOL1 through expression of distinct variants of the main parasite surface antigen Variant Surface Glycoprotein (VSG), respectively termed SRA (Serum Resistance-Associated) and TgsGP (*T. gambiense*-specific Glycoprotein) [9,10]. Whereas SRA inactivates APOL1 following direct interaction with the C-terminal domain of this protein, TgsGP prevents APOL1 activity through increased stiffening of target parasite membranes [8,9,10].

In turn, humans acquired APOL1 C-terminal variants that escape neutralization by SRA and restore resistance to *T. b. rhodesiense* [11]. These APOL1 variants, termed G1 and G2, contain sequence alterations in the C-terminal helix that is targeted by SRA (respectively, S342G/I384M mutations and 388NY389 deletion), which provide resistance to this interaction [11,12]. Moreover, the G1 and G2 variants also affect *T. b. gambiense* infection. Whereas G1 appears to limit infection by *T. b. gambiense*, G2 favors the parasite development [13]. Although still unexplained, this phenotype could result from an eventual difference between APOL1 variant abilities to insert in endosomal membranes of the parasite for trypanosome lysis.

Intriguingly, the G1 or G2 variants are also associated with a higher probability to develop chronic kidney disease, particularly under inflammatory conditions [11,14] and were, therefore, termed “APOL1 risk variants”. This unexpected finding is discussed in Section 3 and Section 4.

### 1.2. The APOL1 Structure

Alternative mRNA splicing generates three main APOL1 isoforms only differing in their N-terminus, two of which are intracellular. Moreover, in the secreted isoform, the signal peptide is cleaved at two alternative sites, generating a protein doublet in the serum. There is no evidence for any differential structure between APOL1 isoforms, but a minor isoform-specific N-terminal peptide exhibits cytotoxic potential [15].

The APOL1 structure is not completely solved but comprises two distinct domains separated by a potential transmembrane (TM) double-stranded helix hairpin (Figure 1).

The N-terminal domain contains five α-helices, with helices 2 to 5 organized as a four-helix bundle [16]. To account for their prominent characteristics, helices 2 and 4 are, respectively, termed here hydrophobic cluster-1 and leucine zipper-1 (HC1 and LZ1). Together with the TM hairpin, this region constitutes a membrane pore-forming domain (PFD), allowing transmembrane ion transport [17,18,19]. Such activity strictly requires acidic conditions, because protonation of negatively charged amino acids in the hairpin helices is absolutely needed for transmembrane insertion of these helices [18,19,20]. Under acidic conditions, APOL1 allows weak anionic fluxes [17], but subsequent medium neutralization triggers high cation conductance [18,21]. APOL1 pore-forming activity is, thus, pH-gated. This gating is controlled by negatively charged residues in the C-terminal domain [22]. Whether these residues belong to the pore is debated [23].

Immediately downstream from the PFD, a membrane-addressing domain (MAD) is required for APOL1 interaction with membranes [17,19,24]. This domain comprises two helices that can fold in a hairpin and contains rows of positively charged amino acids, possibly allowing APOL1 interaction with anionic phospholipids of cellular membranes, such as phosphoinositides and cardiolipin [17,25]. Of particular interest is the presence in MAD of a consensus motif for interaction with cholesterol (cholesterol recognition amino acid consensus, or CRAC: (L/V)-X_1-5_-(Y)-X_1-5_-(K/R)) (268-LAGNTYQLTR-277) [25] (Figure 1). Indeed, no such motif appears to be present in other known APOLs, linking the presence of this CRAC to APOL1 secretion, possibly due to the need for cholesterol binding for HDL-C-mediated transport in the bloodstream.

After a long hinge region, the C-terminal domain contains helices that can strongly interact in cis with helices in the N-terminal region [26,27] (Figure 1). This interaction is induced by pairing between two leucine zipper helices (LZ1 and LZ2 for the N- and C-terminal domain, respectively) and allows mutual masking of the two contiguous hydrophobic helices termed HCs for hydrophobic clusters (Figure 1). Either LZ2 mutations, LZ2 deletion, or acidic pH can disrupt this foldback interaction, increasing HCs exposure, hence APOL1 hydrophobicity [26,27]. Interestingly, HC2 also contains a CRAC motif, termed CRAC-2, shown in Figure 1 (347-LDVVYLVYESK-357). Therefore, through their potential to reduce HC–LZ interaction, LZ2 mutations like G1 or G2 are expected to increase the exposure of CRAC-2, possibly increasing APOL1 interaction with cholesterol. Likewise, APOL1 interaction with cholesterol is expected to be increased in the acidic environment of trypanosome endosomes.

### 1.3. Mechanism of Trypanosome Killing

APOL1 trypanosome killing activity involves HDL-C-driven transport of secreted APOL1 in the blood, followed by APOL1 uptake in the parasite through receptor-mediated HDL-C internalization in the endocytic pathway. This uptake involves another component specific to primate HDL-C, namely Haptoglobin-Related protein (HpR). Indeed, African trypanosomes possess a surface receptor specifically recognizing the Haptoglobin-Hemoglobin (Hp-Hb) complex for efficient capture of the parasite growth factor heme, and this receptor cannot distinguish between Hp-Hb and HpR-Hb [28].

Once in the acidic environment of parasite endosomes, APOL1 undergoes both structure unfolding due to loss of interaction between HC-LZ tandems and insertion into endosomal membranes through acquired transmembrane capacity of the TM hairpin, causing weak chloride fluxes that can trigger osmotic swelling of the parasite lysosome [17]. Endosomal pore-forming activity is not directly responsible for trypanosome killing. It is coupled with both permeabilization of the mitochondrial membrane (MMP, for mitochondrial membrane permeabilization) and an increase in mitochondrial membrane fusion [19]. Only MMP is linked to the parasite death, through MMP-induced release of endonuclease G (TbEndoG) from the mitochondrion inter-membranes space, followed by transfer of this nuclease into the nucleus, causing apoptotic-like DNA degradation [19].

The coupling between endosomal and mitochondrial membrane permeabilization is ensured by the TbKIFC1 kinesin, which is also responsible for intracellular cholesterol traffic [19,29]. Thus, APOL1 association with cholesterol in endosomal membranes, possibly enhanced following CRAC-2 exposure in the acidic medium, probably allows APOL1 transport by TbKIFC1 to the mitochondrion, triggering both apoptotic-like membrane megapore formation and increased membrane fusion. The mechanism involved in these processes is not clear. Trypanosome death may depend on the APOL ability to oligomerize or interact with anti-apoptotic proteins [1,4], but APOL1 also exhibits a membrane fusion potential that depends on membrane insertion under acidic conditions (see Section 2.3). Together with increased fusion of mitochondrial membranes, such potential could be directly or indirectly involved in mitochondrial megapore formation.

Whether TbKIFC1 also transports APOL1-containing endosomes to the trypanosome plasma membrane, thereby inducing APOL1-mediated toxic cation fluxes [18,30] remains unproven. Such cation fluxes, mediated or not by APOL1, could contribute to trypanosome apoptotic-like death, but conversely, they could result from MMP [31].

## 2. Function of the Intracellular APOL1 Isoform(s): Vesicular Trafficking Control

### 2.1. APOL1 Interactions

In podocytes, intracellular APOL1 closely associates with trans-Golgi and endoplasmic reticulum (ER) membranes [24,26]. In immunoprecipitated cellular extracts, the major component associated with APOL1 is Non-muscular Myosin-2A (NM2A), pointing to a function related to membrane dynamics [26]. Accordingly, APOL1 can interact in vitro with the NM2A Regulatory Light Chain (RLC) (E. Pays, unpublished). In addition, APOL1 was found to interact weakly with APOL3 [26,32], and experimental interaction systems also revealed a possible interaction with Prohibitin-2 (PHB2), a mitophagy receptor [27,33]. This interaction was confirmed in vitro [27].

Reciprocally, APOL3 was found to interact weakly with APOL1, but not with PHB2 [27]. Moreover, yeast two-hybrid interaction systems uncovered a specific interaction of APOL3 with Neuronal Calcium Sensor-1 (NCS1), a calcium-dependent activator of the Phosphatidyl Inositol-4-Phosphate [PI(4)P] kinase-B (PI4KB) [26], which synthesizes PI(4)P. An APOL3-specific interaction was also evidenced with PI4KB itself, thus forming a trimeric complex with NCS1. APOL3 can also interact with other PI4KB controlling factors, such as Calneuron-1 (CALN1), a PI4KB inhibitor acting only in the absence of calcium, and ADP-Ribosylation Factor-1 (ARF1), a PI4KB stimulator activated by inflammation-mediated GDP switching to GTP [27]. Interestingly, the interactions of PI4KB and PI4KB controlling factors all occur with the APOL3 HC1 region. In APOL3, such interactions are permitted owing to the absence of LZ2–HC2 foldback interaction with HC1–LZ1 [27]. Conversely, the lack of APOL1 interaction with PI4KB could be ascribed not to sequence divergence, but to the selective foldback folding of APOL1, which impedes the binding of PI4KB to the HC1 region. Indeed, C-terminal deletion was necessary and sufficient to allow strong APOL1 HC1 interaction with PI4KB [27].

### 2.2. Respective Roles of APOL1 and APOL3 in Mitophagy and Apoptosis


*APOL1: Promotion of ATG9A Vesicle Traffic*

*APOL3: Prevention of ATG9A Vesicle Traffic, but Promotion of Mitophagy and Apoptosis Following Inflammation-triggered Traffic*


In human individuals, APOL1 absence due to natural frameshift gene mutations does not cause significant pathology [34,35]. Likewise, in human podocytes cultivated in vitro without specific treatment, experimental APOL1 gene loss induced by genome editing (APOL1 KO) does not lead to any major phenotype [26]. However, under inflammatory conditions induced by the viral mimetic poly(I:C), which activates the interferon type I (IFN-I) signaling pathway, APOL1 KO podocytes exhibit strong inhibition of mitophagy and apoptosis, as compared to WT cells [27]. Consistently, in renal carcinoma cells that are characterized by important inflammation [36], APOL1 KO triggers severe autophagy and mitochondrion dysfunctions [37]. Mitophagy and apoptosis are normally induced by IFN-I through activation of ARF1, which increases ARF1 interaction with PI4KB, linked to Golgi fragmentation [27,38]. The inflammation-linked phenotype of APOL1 KO cells is consistent with the strong APOL1 overexpression that specifically occurs under these conditions [3,4]. Thus, the APOL1 function is clearly linked to membrane dynamics associated with inflammation.

Contrasting with APOL1 KO, APOL3 KO conferred a severe phenotype to cultured podocytes even in absence of poly(I:C)-induced inflammation [26]. This phenotype was characterized by important changes in cell shape and motility, pointing to strong alteration of actomyosin organization. Accordingly, natural APOL3 disruption due to gene mutations was linked to kidney disease, confirming that APOL3 is required for proper podocyte activity [32,39]. In APOL3 KO cells without specific treatment, PI4KB was dissociated from the Golgi, and mitophagy was induced, although incompletely. In addition, upon poly(I:C) treatment, apoptosis of APOL3 KO cells was strongly inhibited, as was also observed with APOL1 KO under the same conditions [26].

Combining APOL1 and APOL3 KOs (APOL1 + 3 KO) did not increase apoptotic death on inflammation but intriguingly led to restoration of Golgi PI4KB activity, associated with abrogation of the APOL3 KO mitophagy phenotype [26]. Thus, APOL1 was clearly needed for both PI4KB dissociation from the Golgi and induction of the APOL3 KO phenotype at the mitochondrion, suggesting a role of APOL1 in PI4KB trafficking to the mitochondrion.

PI4KB traffic from the Golgi to the mitochondrion occurs in vesicles characterized by their association with the lipid scramblase ATG9A (Autophagy-related-protein-9A) [40]. This vesicular traffic, which is linked to Golgi fragmentation, can be induced by IFN-I upon viral infection [38], and conditions mitochondrion membrane fission and the formation of mitochondrial pre-autophagosomal structures (PAS), which assemble at contact sites between the mitochondrion and the ER (MERCs) to initiate mitophagy [40,41,42,43,44] (Figure 2).

Both APOL1 and APOL3 are associated with ATG9A vesicles [27]. Accordingly, APOL1 interacts with the NM2A myosin [26], which is responsible for ATG9A vesicle traffic [45,46,47]. Moreover, APOL1 also interacts with PHB2, a mitophagy receptor that directs vesicular trafficking towards PAS at the mitochondrion [33]. Mitochondrion fission is induced at PAS by PI4KB-mediated synthesis of PI(4)P, which binds to NM2A and membrane-bending factors linked to actin polymerization at the fission site [40,48,49,50]. PI(4)P can oligomerize into lipid microdomains recruiting Golgi-Phosphoprotein-3 (GOLPH3), a trans-Golgi and mitochondrial protein interacting with the NM2A binder Myosin-18A (MYO18) to generate a pulling force for membrane fission and vesicle budding [51,52,53]. Through the control of PI4KB activity, APOL3 is, therefore, indirectly involved in membrane fission, such as that occurring at the mitochondrion for induction of mitophagy [54] (Figure 2). Moreover, APOL3 controls the fusion of mitophagosomes with endolysosomes for mitophagy completion, through interactions between APOL3 helices flanking the TM hairpin and the α-helix of the endosomal Vesicle-Associated Membrane Protein-8 (VAMP8) [27]. Finally, APOL3 also conditions podocyte apoptosis induced by IFN-I [26], likely through its double function in mitochondrial membrane fission and fusion [27,44].

Consistent with the observation that natural APOL3 KO individuals suffer kidney disease [32,39], the cellular phenotype observed experimentally in cultured APOL3 KO podocytes resembled the podocyte phenotype of transgenic mice suffering kidney disease due to G1 or G2 expression. Indeed, this phenotype was characterized by alterations in vesicular trafficking and reduction in autophagic flux [55]. Similarly, the phenotype of natural urine or glomerular G1 or G2 podocytes was characterized by low Golgi PI(4)P levels and Golgi shrinkage, suggesting increased vesicular trafficking from the Golgi [26]. However, as expected from the differential extent of APOL gene editing between these cases, the APOL3 KO phenotype was more severe than those of G1 and G2 and could act in epistasis with APOL1 risk alleles [39].

### 2.3. A Model (Figure 2)

Deletion of either APOL1, APOL3, or both APOLs resulted in complex podocyte phenotypes that can be summarized as follows.
The phenotype of APOL1 KO podocytes is only observed under inflammatory conditions and consists of strong reduction in a process normally induced by IFN-I: Golgi-to-mitochondrion PI4KB trafficking linked to mitophagy and apoptosis.Unlike APOL1 KO, APOL3 KO triggers PI4KB dissociation from the Golgi under non-inflammatory conditions. Thus, APOL3 is clearly required for PI4KB sequestration at the Golgi. However, the induction of mitophagy and apoptosis linked to APOL3 KO-mediated PI4KB traffic results in abortive mitochondrial membrane fission and fusion. Thus, APOL3 must also play a role in these processes.In APOL1 + APOL3 double KO podocytes, the APOL3 KO mitochondrial phenotype (induction of abortive mitophagy and apoptosis irrespective of inflammation) is totally absent, indicating that APOL1 is responsible for allowing this phenotype. Given the key role of PI4KB for mitochondrion fission and mitophagy [40,41,44], a straightforward explanation is that APOL1 conditions the PI4KB transfer from the Golgi to the mitochondrion. Such function is compatible with the activities of the APOL1 partners: NM2A for the traffic of PI4KB-carrying ATG9A vesicles and PHB2 for vesicular targeting to the mitochondrion.APOL1 KO, APOL3 KO, and APOL1 + 3 KO all induce a similar level of inflammation-linked apoptosis, suggesting complementary activities of APOL1 and APOL3: APOL1 ensures the traffic of APOL3-carrying vesicles for induction of apoptosis, and APOL3 is involved in apoptosis.The specific role of APOL3 in both mitophagy and apoptosis can readily be explained by the specific ability of APOL3 to perform transmembrane insertion at neutral pH [19,20,27] and by the specific interaction of APOL3 with the fission factor PI4KB and the fusion factor VAMP8, strikingly contrasting with the APOL1 inability to perform both activities [26,27,44]. Such inability results from the strict APOL1 dependence on acidic conditions for insertion into membranes, because preincubation of APOL1-containing vesicles under acidic conditions was necessary and sufficient to confer to these vesicles the ability for fusion with VAMP8-containing vesicles [27]. In this respect, it is worth emphasizing the key pH difference between the cellular environments of intracellular APOL1 in podocytes and trypanosomes, which reflect opposite trafficking pathways (respectively secretion and uptake) and must result in distinct APOL1 activities given the pH sensitivity of this protein [23].

## 3. Effects of Intracellular APOL1 Risk Variants: Interference with APOL3 Control of Membrane Dynamics

### 3.1. Indirect Effects on Mitophagy and Apoptosis

Deletion of the C-terminal APOL1 helix at V353 (APOL1Δ) mimicked the effects of APOL3 KO, leading to a phenotype also resembling that of G1 or G2 podocytes but with increased severity [26,27]. A common feature of the APOL1Δ, G1, and G2 variants is disruption of cis-interaction between the C- and N-terminal LZ helices, resulting in increased HC exposure and, therefore, increased interaction between the APOL1 and APOL3 HCs, which inactivates APOL3 [26]. As expected from their relative disruptive effect on interaction between LZs, point mutations in LZ2 (as in G1 and G2) triggered a milder phenotype than LZ2 deletion (as in APOL1Δ) [26].

Thus, like occurs following APOL3 KO, APOL3 inactivation due to interaction with APOL1 C-terminal variants can induce the release of PI4KB from sequestration at the Golgi and PI4KB transfer to the mitochondrion, mimicking the effects of IFN-I activation but without activation of ARF1. Abortive mitochondrion fission observed in either APOL3 KO or APOL1Δ cells irrespective of inflammation [27] can be explained by insufficient PI4KB activity because PI4KB-mediated fission of the mitochondrion requires ARF1 activation [41,56], which does not seem to be triggered following APOL3 KO or APOL1Δ expression. While impairment of fission results in inhibition of mitophagosome formation, loss of APOL3 in APOL3 KO cells, or APOL3 inactivation in APOL1Δ cells, also prevents the APOL3 activity in mediating the fusion between mitophagosomes and endolysosomes and, thus, inhibits the completion of mitophagy [27,44]. Therefore, intracellular APOL1 risk variants are expected to induce abortive mitophagy and apoptosis, even under non-inflammatory conditions. However, such effects should be milder than those of APOL1Δ.

### 3.2. No Evidence for Ion Pore-Forming Activity Within the Podocyte

In trypanosome endosomes or in synthetic membranes in vitro, APOL1 exhibits ion pore-forming activity that requires acidic conditions. In podocytes, the secreted APOL1 risk variant isoforms must clearly avoid intracellular membrane insertion to allow their extracellular release. Such must also be the case of intracellular risk variant isoforms, even under inflammatory conditions. Indeed, these intracellular isoforms were shown to be devoid of any cytotoxicity, in clear contrast with the extracellular isoforms [57,58,59]. Therefore, intracellular trafficking of both WT and variant APOL1 does not encounter sufficiently acidic conditions for transmembrane insertion.

### 3.3. Possible Indirect Effects on Cation Fluxes

As a result of APOL1 variants-mediated PI4KB and NCS1 release from sequestration at the trans-Golgi membrane [26,27], PI4KB and NCS1 delocalization could affect the activity of components known to be controlled by these proteins. Such is particularly the case of infection-mediated inflammation, which triggers Golgi fragmentation and vesicular trafficking from the Golgi [38]. PI4KB and NCS1 can delocalize to the mitochondrion, associated with induction of mitochondrion fission and mitophagy [40,41,54]. However, PI4KB delocalization may also occur to endosomes along with activation of the inflammatory Stimulator of Interferon Genes (STING) [60], which depends on STING interaction with PI(4)P [61], and to the plasma membrane for phagocytosis linked to pathogen infection [62]. The transfer of PI4KB to the plasma membrane could affect cellular signaling induced by the phosphoinositide PI(4,5)P2, which influences the activity of different cation channels [63,64,65,66,67,68]. In addition, NCS1 transfer to MERCs could promote ER-to-mitochondrion calcium fluxes through IP3-Receptors (IP3R) [69,70], and NCS1 transfer to the plasma membrane could activate G-coupled receptors and consecutive cellular signaling [71,72,73,74,75]. Thus, cation fluxes are expected to be altered following PI4KB and/or NCS1 delocalization, possibly causing pleiotropic changes in actomyosin organization and activation of stress processes.

## 4. Effects of Extracellular APOL1 Risk Variants: Plasma Membrane Disturbance

### 4.1. No Evidence for Ion Pore-Forming Activity at the Cell Surface

In multiple in vitro studies involving diverse cell types including human podocytes, important cytotoxicity was observed following ectopic expression of the APOL1 variants G1 or G2, but not WT APOL1 [37,57,58,59,76,77,78]. Among possible artifacts linked to this type of experimental studies, ectopic gene expression is often linked to a bias for increased activity, and the cell culture medium significantly differs from the natural environment of secreted APOL1 in the blood, particularly regarding the amount of the APOL1 carrier HDL-C, which is much lower in vitro [23,25].

In vivo, specific podocyte cytotoxicity of the APOL1 risk variants was linked to the strong induction of APOL1 expression occurring during infection-induced inflammation, particularly with viruses [78,79,80,81], and it was ascribed to the circulating isoform of the APOL1 variants [58,78]. Accordingly, the major APOL1 cytopathology was clearly shown to result from interaction of extracellular APOL1 variants with the podocyte surface, whereas intracellular APOL1 isoforms were not toxic [24,57,58,59].

The exact nature of APOL1 variant interaction with the podocyte surface is not known, but it is generally considered as cation pore formation following transmembrane insertion of APOL1, because APOL1 variants induce cation fluxes both in cells cultured in vitro and under inflammatory conditions in vivo [82,83,84,85]. Accordingly, in vitro studies with synthetic membranes revealed increased cation channel activity of APOL1 variants as compared with the WT form [86]. Targeting surface APOL1 by different tools, such as specific anti-APOL1 antibodies, *T. b. rhodesiense* SRA or the specific drug inaxaplin, resulted in loss of APOL1 toxicity, interpreted as loss of APOL1 pore-forming activity [82,83,84,86,87]. This interpretation resulted not from the demonstration of such activity, but from the comparison with a model for APOL1-mediated trypanosome lysis, which invokes the key role of cation surface fluxes induced by APOL1 [18,30].

However, transmembrane insertion of the APOL1 PFD in the extracellular environment is very unlikely given its strict acidic pH-dependence, and such insertion has not been demonstrated. Moreover, the increased level of channel activity observed in vitro with the APOL1 variants compared with WT APOL1 [86] cannot fully explain the difference in APOL1-induced cytotoxicity observed in vivo. Finally, and importantly, only the MAD, and not the PFD, was found to be inaccessible to antibodies at the podocyte surface [24], suggesting deep interaction of only this domain with the plasma membrane, which rules out any possibility of membrane insertion of the PFD for APOL1 pore-forming activity.

The phenotype of APOL3 KO or APOL1Δ podocytes provides a clear confirmation that APOL1 nephropathy can occur independently from APOL1 cation pore-forming activity. Both phenotypes resemble those of G1 or G2 but with increased severity that can be explained by a stronger APOL3 inactivation [26,27]. Whereas APOL3 KO cannot account for eventual APOL1 pore-forming activity, APOL1Δ is devoid of such activity [18]. Thus, the G1/G2-like phenotype induced in either case cannot result from the formation of APOL1 cation pores.

In conclusion, the effect of APOL1 variants on cation fluxes, considered responsible for the major cytopathology induced by these variants either in cultured cells in vitro or upon infection-linked inflammation in vivo, is unlikely to result from the formation of APOL1 channels at the podocyte surface but more likely reflects the activation of other channels in the podocyte plasma membrane.

### 4.2. Effects on Plasma Membrane Organization: Interaction with Cholesterol

In the serum, secreted APOL1 is tightly bound to cholesterol-rich HDL-C particles, and in trypanosomes, secreted APOL1 is trafficked by the TbKIFC1 kinesin, which carries cholesterol-containing membranes [2,19,29] (Figure 3). Thus, APOL1 may bind to cholesterol. Accordingly, APOL1 exhibits one or two cholesterol-binding consensus motifs (CRACs), depending on LZ2-mediated protein folding. Therefore, it can be envisaged that when not sequestered by HDL-C particles, secreted APOL1, whether WT or variant, interacts with cholesterol at the podocyte surface. In such a case, APOL1 risk variants could interact more strongly than WT APOL1, given the exposure of CRAC-2 in the variants only.

A strong argument for the involvement of cholesterol in podocyte cytotoxicity by APOL1 is the phenotype of the N264K APOL1 variant. This mutation abrogates G1 or G2 cytotoxicity, despite a location outside the PFD [95,96]. Interestingly, N264 is in the MAD, very close to CRAC-1, and modeling predicts that N264K should interfere with CRAC-1 binding to cholesterol because such interaction necessitates the deep penetration of N264 into the upper membrane layer, which is expected to be prevented following N264K mutation [25]. Consistent with the role of cholesterol in intracellular trafficking of APOL1 to the mitochondrion, the N264K APOL1 variant exhibits slightly reduced trypanosome killing potential in vitro [97]. However, the important N264K variant pore-forming activity still exerted in trypanosomes under these conditions [97] clearly contrasts with the abrogation of cytotoxicity in podocytes [95,96], further suggesting that podocyte cytotoxicity does not result from APOL1 variant pore-forming activity.

Similarly, C-terminal APOL1 truncation at V353 (in APOL1Δ), which disrupts CRAC-2, also results in loss of APOL1 variant cytotoxicity [57,58]. This is especially remarkable, given the protein unfolding of this variant, which allows increased hydrophobicity linked to HCs exposure [26,27]. Thus, the cytotoxicity of APOL1 C-terminal variants does not simply result from increased hydrophobicity, but could more specifically be due to CRAC-2 exposure for interaction with cholesterol.

Finally, interaction of APOL1 variants with lipid droplets, which contain free cholesterol like HDL-C particles, also results in reduction in APOL1 variants cytotoxicity, further supporting the role of cholesterol interaction in this process [98].

In conclusion, APOL1 variant cytotoxicity on podocytes likely results from interaction of APOL1 with cholesterol at the cell surface, rather than from APOL1 pore-forming activity. This can only occur under high APOL1/low HDL-C conditions, which allow APOL1 availability for such interactions. Given that CRAC-1 and CRAC-2 appear to be simultaneously required for toxicity, the lack of toxicity by WT APOL1 probably results from CRAC-2 inaccessibility due to LZ2-mediated protein folding. The need for a combination of cholesterol binding at two distant sites suggests the involvement of cholesterol clustering in the APOL1 variants toxic effects.

Importantly, these considerations are the first to provide a straightforward explanation for the key role of cholesterol in chronic kidney disease [91,92,93,94,99].

### 4.3. Putative Effects on Cation Channels: Alteration of Lipid Microdomains

If not due to APOL1 cation pore-forming potential, the induction of toxic cation fluxes by APOL1 variants should occur through activation of podocyte cation channels. Interestingly, two major cation channels are crucially involved in podocyte architecture and activity, namely the Ca^2+^ Transient Receptor Potential Cation Channel-6 (TRPC6) [100,101,102,103,104] and the Ca^2+^-dependent Big K+ conductance (BK) channel [105,106]. The activity of these channels is controlled by both cholesterol and PI(4,5)P2 [107,108,109,110,111,112,113,114], present at the outer and inner layers of the plasma membrane, respectively, and associated in lipid microdomains by transmembrane coupling [115] (Figure 3).

Cholesterol recruits podocin, which oligomerizes in pedicels lipid rafts together with nephrin to form the filtration slits. Podocin also recruits cholesterol to the TRPC6 ion channel complex at the slit diaphragm, organizing ion channel–lipid complexes that are very sensitive to mechanical tension on these microdomains [116,117,118,119]. Mechanical tension on cholesterol microdomains involves the actin cytoskeleton [120] and influences TRPC6 and BK channels conductance together with associated IP3R receptors [107,120,121], conditioning podocyte activity [110,118]. Therefore, it is tempting to speculate that increased APOL1 variant interaction with cholesterol, which involves two distant CRAC motifs, affects the clustering or tension of the lipid rafts, activating toxic cation fluxes (Figure 3). Consistently, the higher hydrophobicity of the G1 and G2 variants [26] is linked to higher APOL1 clustering in microdomains at the podocyte surface [59]. Increased hydrophobicity of APOL1 variants probably also accounts for their facilitated cation conductance in synthetic membranes in vitro [86], as well as for their ability to increase cholesterol accumulation in both HDL-C particles and tissues [122,123].

### 4.4. The Illuminating Case of APOL1Δ

In trypanosomes, APOL1Δ does not exhibit any toxic activity [12], in keeping with its inability to form ion pores [18]. This confirms that transmembrane insertion is involved in trypanosome killing by APOL1 [19,20]. In podocytes, APOL1Δ induces a severe G1- or G2-like phenotype [26,27], indicating that the G1/G2 phenotype does not result from APOL1 pore-forming activity. Finally, in experimental cellular assay systems, APOL1Δ is totally devoid of cytotoxicity, in clear contrast with G1 or G2 [57,58], indicating that APOL1 variant-specific cytotoxicity can be uncoupled from the main G1/G2 phenotype. In APOL1Δ, not only the PFD but also the CRAC-2 motif, is not functional. Thus, either the PFD or CRAC-2 could account for APOL1 variant-specific cytotoxicity. However, contrasting with the PFD, only CRAC-2 is expected to be specifically active in G1 or G2, and not in the WT [26,27]. Thus, G1- or G2-specific cytotoxicity likely results from CRAC-2 cholesterol-binding activity, as also supported by other arguments listed above.

## 5. Conclusion: Evolution of the APOL1 System of Innate Immunity Against African Trypanosomes

The molecular “arms race” between African primates and African trypanosomes involves successive steps that illustrate the continuous and dynamic interplay between host and parasite (Figure 4).

•Step 1: the invention of a specific toxin systemFrom intracellular functions linked to membrane dynamics (membrane traffic, fission and fusion), which control several infection-linked processes such as vesicular trafficking, apoptosis, and mitophagy as well as membrane disruption of intracellular bacteria [124], the APOL family of African primates evolved to include a novel function by a secreted member, APOL1, to kill the deadly bloodstream parasite *T. brucei*, responsible for the devastating sleeping sickness disease in humans and nagana in cattle [7]. To reach the circulating trypanosomes, APOL1 was provided with a cholesterol binding motif for anchoring to circulating HDL-C particles, normally involved in cholesterol recycling. Moreover, these HDL particles were associated with another primate-specific protein, HpR, recognized as growth factor by the parasite, hence promoting the uptake of APOL1 in the parasite endosomes (akin the Trojan horse). In the acidic environment of endosomes, APOL1 can insert into the vesicular membrane, and a second cholesterol-binding motif becomes accessible, possibly favoring intracellular trafficking of APOL1-containing endosomes by the cholesterol-carrying TbKIFC1 kinesin. APOL1 is trafficked to the mitochondrion and directly or indirectly triggers megapore formation in the outer mitochondrial membrane, causing apoptotic-like death of the parasite. Besides the HDL-C role in APOL1 circulation and uptake, HDL-C-mediated sequestration of APOL1 also prevents any eventual toxicity of this protein in the blood, which would possibly result from interaction with cholesterol at the surface of kidney podocytes.•Step 2: the parasite reaction: different APOL1 resistance factorsFrom its role in antigenic variation for resistance to the mammalian immune response, the dynamic trypanosome genetic system that controls the expression and evolution of Variant Surface Glycoprotein (VSG) gene families, invented two VSG-derived resistance factors to avoid APOL1 killing activity [8]. Given their key involvement in human infectivity, these factors characterize the two *T. brucei* subspecies *T. b. rhodesiense* and *T. b. gambiense*, widespread in eastern and western Africa, respectively [125,126]. Whereas *T. b. rhodesiense* SRA inactivates APOL1 by direct interaction, *T. b. gambiense* TgsGP prevents APOL1 toxic activity through increased stiffening of target parasite membranes.•Step 3: APOL1 countermeasures: more trypanolytic, but potentially nephrotoxic, APOL1 C-terminal variantsThe C-terminal APOL1 variants G1 and G2 can escape SRA interaction, allowing human resistance to *T. b. rhodesiense* infection. G1 and G2 spreading in western Africa probably led to the disappearance of *T. b. rhodesiense* from this part of the continent, whereas *T. b. gambiense* was allowed to take over despite some resistance by the G1 variant. Although this limiting effect of G1 on *T. b. gambiense* growth is still unclear, the hypothesis that C-terminal APOL1 variants cause cellular toxicity through cell surface interaction with cholesterol opens the perspective of evolutive selection of some APOL1 variants through acquired activity against other bloodstream pathogens [127]. However, the improved resistance to trypanosomes was associated with a price to pay. The G1- and G2-bearing individuals exhibit high probability of chronic kidney disease, particularly upon infection by viruses. In this review, I argue that the major cytotoxicity of APOL1 variants results from their increased interaction with cholesterol at the podocyte surface, due to structure unfolding that allows the cholesterol-binding activity of the CRAC-2 motif. This interaction triggers activation of cation fluxes by cholesterol-sensitive channels normally controlling podocyte filtration activity. APOL1 availability for such interaction may result from the simultaneous APOL1 increase and HDL-C decrease induced by inflammation. The reason for the preferential effects of APOL1 risk variants on kidney podocytes could be the particular importance of surface cholesterol rafts for the structure and filtration function of the slit diaphragm. Through interactions with both lipids and actomyosin, APOL1 can crucially affect the function of these structures.•Step 4: towards improved safety?The recent finding that the N264K mutation abrogates G1 or G2 toxicity suggests an evolutionary trend in the APOL1 system towards improved safety, through reduced CRAC-1 interaction with cholesterol. As a corollary, the in vitro trypanosome killing potential of this mutant is slightly reduced. However, CRAC-1 is probably mainly responsible for the association of secreted APOL1 with HDL-C particles in the bloodstream, whereas CRAC-2, only accessible under acidic conditions, could be particularly involved in APOL1 trafficking within the trypanosome. Therefore, CRAC-1 inactivation by N264K could principally affect the delivery of APOL1 to the parasite in the blood. Such reduction in trypanosome killing capacity is expected to be largely harmless out of Africa, where only some parasites related to African trypanosomes, like *T. evansi*, are present. As exemplified by the exceptional case of trypanosome infection in India [34], even the complete loss of APOL1 does not appear to induce significant health problems, apart from sensitivity to *T. evansi*. However, the absence or inactivation of APOL1, such occurs with inaxaplin treatment [87], is expected to impair the induction of mitophagy under inflammatory conditions, which should aggravate the pathology of viral infection. In support to this conclusion, deletion of APOL1 in renal carcinoma cells, which are characterized by inflammation, results in severe mitochondrial dysfunctions [37].

## Figures and Tables

**Figure 1 cells-13-01738-f001:**
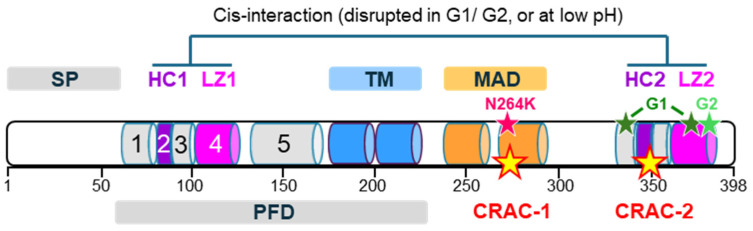
Scheme of APOL1 structure (398 amino acids). The different colored cylinders are different α-helices. SP = signal peptide; HC = hydrophobic cluster; LZ = leucine zipper; TM = transmembrane span; MAD = membrane-addressing domain; PFD = pore-forming domain; CRAC = cholesterol recognition amino acid consensus. The approximate locations of the N264K, G1, and G2 mutations are indicated by pink, dark green, and light green stars, respectively, and the CRAC motifs are indicated by yellow stars.

**Figure 2 cells-13-01738-f002:**
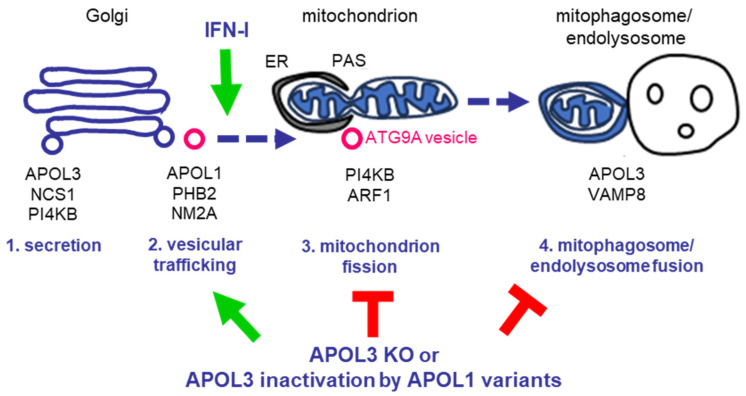
APOL1 involvement in membrane dynamics [26,27,44]. Under non-inflammatory conditions (1), APOL1 is associated with APOL3 at trans-Golgi and endoplasmic reticulum membranes. APOL3 controls PI4KB activity for vesicular secretion in a trimeric complex with NCS1. Upon infection-induced inflammation (2), IFN-I signaling triggers ARF1 activation and dissociation of PI4KB from the Golgi in Golgi-derived ATG9A vesicles also carrying APOL1, APOL3, PHB2, and NM2A. While NM2A drives vesicular trafficking, the mitophagy receptor PHB2 targets the vesicles to mitochondrial pre-autophagosomal structures (PAS) at mitochondrion–endoplasmic reticulum (ER) contact sites. PI4KB and ARF1 induce mitochondrial membrane fission, releasing mitophagosomes (3). The mitophagosome fusion with endolysosomes (4) allows completion of mitophagy. This fusion is triggered by interactions between APOL3 and endosomal VAMP8. At each step, relevant key protein complexes are mentioned. Either APOL3 KO or APOL3 inactivation by APOL1 C-terminal variants leads to PI4KB dissociation from the Golgi in a process mimicking that induced by IFN-I, except that the absence of ARF1 activation and lack of APOL3 activity result in defective mitophagy (incomplete membrane fission and fusion). In addition to the induction of mitophagy, APOL1 trafficking activity to the mitochondrion also conditions APOL3-mediated apoptosis induced under inflammatory conditions.

**Figure 3 cells-13-01738-f003:**
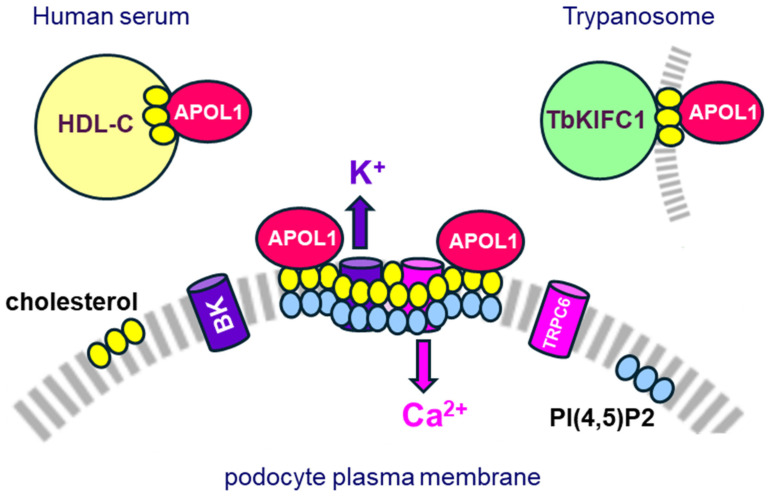
The role of cholesterol in APOL1 trafficking and toxic activity (adapted from [25]). APOL1 is associated with the serum cholesterol carrier HDL-C or with the *T. brucei* cholesterol carrier TbKIFC1. At the surface of kidney podocytes, the activity of Ca^2+^ TRPC6 channels and Ca^2+^-dependent K^+^ BK channels is controlled in cholesterol microdomains involving the cholesterol-binding podocin and the actin cytoskeleton. Through interaction with cholesterol in the upper layer of the podocyte plasma membrane [25], APOL1 variants may increase cation fluxes by TRPC6 and BK channels, causing cytotoxicity. Normally, secreted APOL1 is sequestered by HDL-C particles, restraining the availability of APOL1 for interaction with the podocyte surface. However, such limitation is lost under two types of situations: in cell culture in vitro, or upon inflammation in vivo, such occurs upon virus infection. In both cases, APOL1 synthesis is increased whereas the HDL-C amount is reduced [3,4,88,89,90], increasing APOL1 availability for toxic surface interaction. Accordingly, low HDL cholesterol is a known predictor of chronic kidney disease [91,92,93,94].

**Figure 4 cells-13-01738-f004:**
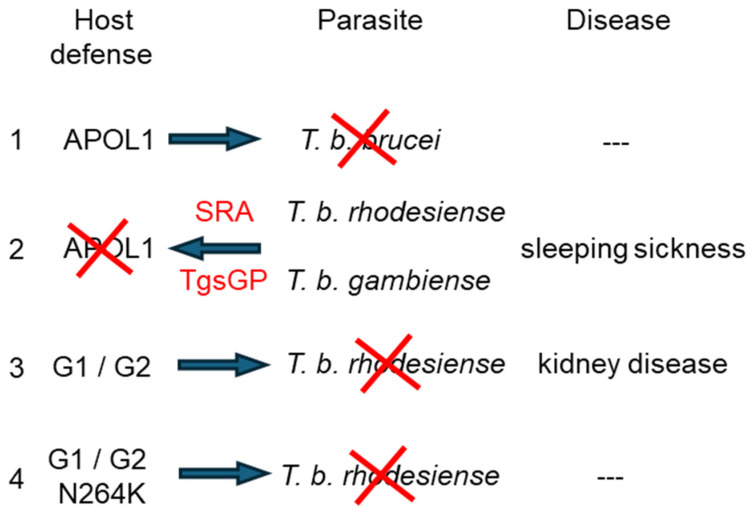
Molecular ‘arms race’ between African trypanosomes and humans. (1) Human ancestors produced a secreted version of APOL1 to kill the extracellular parasite *Trypanosoma brucei brucei*. (2) Two *T. b. brucei* clones, known as *T. b. rhodesiense* and *T. b. gambiense*, used variant surface glycoprotein (VSG)-derived proteins termed SRA and TgsGP, respectively, to neutralize APOL1. (3) The APOL1 C-terminal variants G1 and G2 allowed human resistance to SRA, hence human ability to kill *T. b. rhodesiense*. However, this also resulted in a high probability of humans developing chronic kidney disease. (4) C-terminal APOL1 variants with the N264K mutation can still efficiently lyze *T. b. rhodesiense*, but no longer exhibit kidney cytotoxicity.

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
