# Peer review of "Apolipoprotein-L1 (APOL1): From Sleeping Sickness to Kidney Disease"

_cells, 2024, doi:10.3390/cells13201738_

Round 1
Reviewer 1 Report
Comments and Suggestions for Authors
The manuscript by Etienne Pays presents a comprehensive review discussing the impact of APOL1 unfolding caused by G1 or G2 mutations, which leads to podocyte dysfunction and impairs renal filtration. This well-structured study highlights how APOL1 variants induce inflammation-associated podocyte toxicity through disturbances in the plasma membrane, primarily due to enhanced interaction with cholesterol, which in turn increases cation channel activity rather than pore formation.
However, the following points require revisions from the author:
1. Please emphasize the unique aspects of this review to differentiate it from similar works, such as “doi: 10.7554/eLife.29056.” Highlighting how this review provides new insights or focuses on different mechanisms, such as the interaction between APOL1 variants and cholesterol-mediated podocyte dysfunction, will help underline its novel contributions.
2. In line 41, the term "invented" should be replaced with "discovered" when referring to the APOL1 isoform, as it reflects a more accurate description of its identification.
3. In line 60, when discussing the role of APOL1 variants G1 or G2 in the development of chronic kidney disease (CKD), the author could enhance the discussion by considering the potential effects of APOE and the APOE-4 allele in inflammation-related CKD. A relevant study to cite could be “doi: 10.1093/ckj/sfae294”.
4. Please discuss and explain the term “Sleeping sickness” in the introduction. It appears in the list of keywords but is not used in the manuscript. It would be better if the authors could incorporate this term when introducing Trypanosoma brucei for the first time and again in the conclusion section, providing a clear link between the disease and the parasitic organism discussed.
5. In figure 1 legend, please correct the terminology "CRAC motives", to "motifs.".
6. Please provide the full form of the acronym “ARF1” in line 187.
7. Please rephrase line 331 to avoid starting with "because." Consider rewording it to something like: "As a result of these findings, it is evident that..." to improve the flow and readability.
8. In line 435, the concluding sentence is unclear. Please revise it to more clearly convey the author's intended message.
9. It would be better if the author could possibly include a summary flowchart that depicts step-by-step evolution of the APOL1 system in innate immunity against African trypanosomes. This would provide readers with a clearer visual understanding of the processes discussed.
Reviewer 2 Report
Comments and Suggestions for Authors
Since its discovery in the late 1990s, numerous studies have highlighted the implications of apolipoprotein L1 (APOL1) in adaptation to parasitic diseases and the pathogenesis of kidney disease. In particular, several mechanisms of APOL1 cellular toxicity have been proposed, which may underlie the association between APOL1 risk variants and kidney disease. In this respect, it has been reported that APOL1 acts as an ion channel in the lysosomal membranes of human and parasitic cells, causing cell swelling and death. In this study, the author very nicely argues that APOL1 variants induce inflammation-related toxicity in podocytes not through pore formation, but through disruption of plasma membranes resulting from increased interaction with cholesterol, which enhances cation channel activity. A naturally occurring mutation in the membrane interaction domain cancels out the toxicity of the APOL1 variant at the cost of increased sensitivity to trypanosomes, illustrating the ongoing mutual adaptation between host and parasite.
This is an excellent and highly original review on apolipoprotein L1. The manuscript is very well written and therefore it is acceptable in its current version.
Reviewer 3 Report
Comments and Suggestions for Authors
I have no additional comments or revisions to suggest for this review article. It is comprehensive, insightful, and well-written, offering a valuable synthesis of the current understanding of APOL1’s role in immunity and kidney disease. Dr. Pays' research legacy in this area is well-recognized, and this review is another excellent contribution to the field. I thank the author for this thorough and thought-provoking work, and it has been a pleasure to review this review article.
